# Fresh Medium or L-Cystine as an Effective Nrf2 Inducer for Cytoprotection in Cell Culture

**DOI:** 10.3390/cells12020291

**Published:** 2023-01-12

**Authors:** Wujing Dai, Qin M. Chen

**Affiliations:** Department of Pharmacy Practice and Science, College of Pharmacy, University of Arizona, 1295 N Martin Ave, Tucson, AZ 85721, USA

**Keywords:** amino acid, cytoprotection, non-toxic, transcription, antioxidant and detoxification genes, oxidative stress

## Abstract

The Nrf2 gene encodes a transcription factor best known for regulating the expression of antioxidant and detoxification genes. A long list of small molecules has been reported to induce Nrf2 protein via Keap1 oxidation or alkylation. Many of these Nrf2 inducers exhibit off-target or toxic effects due to their nature as electrophiles. In searching for non-toxic Nrf2 inducers, we found that a culture medium change to fresh DMEM is capable of inducing Nrf2 protein in HeLa, HEK293, AC16 and MCF7 cells. Testing the components of DMEM led to the discovery of L-Cystine as an effective Nrf2 inducer. L-Cystine induces a dose-dependent increase of Nrf2 protein, from 0.1 to 1.6 mM. RNA-seq analyses and RT-PCR revealed an induction of multiple Nrf2 downstream genes, including NQO1, HMOX1, GCLC, GCLM, SRXN1, TXNRD1, AKR1C and OSGIN1 by 0.8 mM L-Cystine. The induction of Nrf2 protein was dependent on L-Cystine entering cells via the cystine/glutamate antiporter and the presence of Keap1. The half-life of Nrf2 protein increased from 19.4 min to 30.9 min with 0.8 mM L-Cystine treatment. L-Cystine was capable of eliciting cytoprotection by reducing ROS generation and protecting against oxidant- or doxorubicin-induced apoptosis. As an amino acid derivative, L-Cystine is considered a non-toxic Nrf2 inducer that exhibits the potential for protection against oxidative stress and tissue injury.

## 1. Introduction

The Nrf2 gene has attracted intense interest since its discovery in 1994 due to its ability to elicit a myriad of cytoprotective responses [1,2,3,4,5,6]. This gene encodes a transcription factor best known for regulating the expression of genes containing the Antioxidant Response Element (ARE) in the promoters [7,8]. A long list of antioxidant and detoxification genes are downstream targets of Nrf2 [5,6]. Typical examples include those encoding key components of redox systems, such as NAD(P)H quinone oxidoreductase 1 (NQO1), heme oxygenase 1 (HMOX1), glutamate-cysteine ligase catalytic subunit or regulatory subunit (GCLC, GCLM), glutathione peroxidases, glutathione reductase, peroxiredoxin, sulfiredoxin 1 (SRXN1), thioredoxin and thioredoxin reductase 1 (TXNRD1). Genomic or transcriptomic profiling has led to the discovery of Nrf2 impact on multiple cellular and molecular pathways, from autophagy to mitochondrial turnover, cell proliferation or extracellular remodeling [5,6]. These findings are consistent with the observations that knocking out Nrf2 leads to an increased susceptibility of mice to a variety of physical or chemical stresses and disease inducers.

Oxidative stress causes the activation of Nrf2 due to increased protein stability, nuclear translocation and de novo protein translation [5,6]. Nrf2 protein normally binds to Keap1 in the cytoplasm, where Keap1 recruits ubiquitin ligases for Nrf2 ubiquitination, contributing to a short half-life of Nrf2 protein, 10–20 min. When Keap1 is modified by oxidation or alkylation, or removed due to autophagy, Nrf2 is relieved from ubiquitination, causing an increase in Nrf2 protein half-life to 40–60 min [3,4,9,10,11]. In addition to Keap1, two additional mechanisms, HMG-CoA Reductase Degradation protein 1 (Hrd1), also known as Synoviolin, and β-transducin repeat-containing protein (β-TrCP), can carry out Nrf2 protein ubiquitination. Hrd1/Synoviolin responds to ER stress, whereas β-TrCP is activated downstream of GSK3β activity [2,12]. Recently, we and others have found that Nrf2 is among a few genes that undergo de novo protein translation when mammalian cells encounter acute oxidative stress [13,14,15,16,17,18]. De novo Nrf2 protein translation may cooperate with Nrf2 protein stabilization for the ultimate activation of Nrf2 as a transcription factor.

A number of small molecules have been developed for Nrf2 induction. Many of these molecules alkylate or oxidize Keap1, therefore removing the negative force of Keap1 on Nrf2 [4,19,20]. The most commonly studied Nrf2 inducers include sulforaphane (SFN) and bardoxolone methyl (CDDO-Me). In addition, multiple small molecules derived from herb medicine, spices and dietary plants are capable of inducing Nrf2 [5,9,21,22]. The chemical nature of these compounds as electrophiles suggests off-target effects and toxicity. Given these caveats, it is imperative to identify Nrf2 inducers that can bypass the issue of toxicity.

In an effort of searching for non-toxic Nrf2 inducers, we tested the components in a cell culture medium. Several growth factors have been reported to induce oxidative stress. Examples include Epidermal Growth Factor, Insulin-like Growth Factor-1, Transforming Growth Factor beta 1 and Vascular Endothelial Growth Factors [23,24,25]. Given the non-toxic nature of growth factors, we asked whether growth factors could induce Nrf2. With cell culture systems, we tested whether the replenishment of a fresh culture medium with the growth factor-containing serum induced Nrf2. Although fetal bovine serum did not induce Nrf2, we found that an amino acid derivative in a cell culture medium, i.e., L-Cystine, is an effective Nrf2 inducer.

## 2. Materials and Methods

Cell culture: HeLa (CCL-2) cells were obtained from ATCC. AC16 human cardiomyocytes (SCC109) were purchased from Millipore Sigma. HEK293 and MCF7 cells were obtained from Dr. Donna Zhang or Andrew Paek’s laboratory at the University of Arizona. Cells were cultured in high-glucose Dulbecco’s Modified Eagle’s medium (DMEM) with 10% fetal bovine serum (FBS), 100 U/mL penicillin and 100 μg/mL streptomycin at 37 °C with 5% CO_2_. Cells at passage 4 to 20 were seeded in tissue culture plates and treated with L-Cystine when the confluence reached 80%. L-Cystine (Research Products International, Mount Prospect, IL, USA) was dissolved in a 500 mM HCl or 200 mM NaOH solution to make a 0.1 M stock solution for its addition to cells at the final concentration of 0.1 to 1.6 mM.

Western blot to detect protein level changes: Cells in culture dishes were harvested in a 1× Laemmli sample buffer containing 5% 2-mercaptoethanol. After boiling and sonicating, samples with an equal cell number or protein concentration were loaded onto a gel for SDS-PAGE. Proteins on the gel were transferred to a PVDF membrane using the Bio-Rad (Hercules, CA, USA) Trans-Blot Turbo Transfer System and incubated with the following antibodies: anti-Nrf2 (sc-13032), anti-Keap1 (sc-365626), anti-HMOX1 (sc-136960), anti-SRXN1 (sc-514940), anti-TXNRD1 (sc-28321), anti-AKR1C (sc-166297), anti-GAPDH (sc-32233) or anti-Ubiquitin (sc-8017), which were purchased from Santa Cruz Biotechnology. Anti-caspase-3 (#9662) was purchased from Cell Signaling Technology (Danvers, MA, USA) for Western blot. The bound antibodies were recognized by anti-mouse IgG-HRP (A9044-2ML) or anti-rabbit IgG-HRP (A9169-2ML), which were purchased from Sigma Aldrich (St. Louis, MO, USA).

RNA-seq for transcriptomics: Total RNAs were extracted from 3 independent sample preparations with control or 0.8 mM L-Cystine-treated for 16 h by RNeasy Kits (Qiagen, Germantown, MD, USA). Sample quality control, transcriptome library preparation, library quality control, sequencing, raw data output, data quality control and bioinformatics analysis were performed by BGI Genomics Americas Corporation (San Jose, CA, USA). Briefly, raw reads were filtered to remove the reads containing the adaptor sequence, unknown nucleotides greater than 5%, or a quality score less than 15 by SOAPnuke software [26]. The clean reads were aligned to a reference genome GRCh38.p12 using HISAT [27] or to a reference transcriptome using Bowtie 2 [28]. RSEM was used to calculate the gene expression level of each sample [29]. For detecting differential genes, the DESeq2 method was employed with Qvalue (Adjusted Pvalue) ≤ 0.05 [29].

RT-qPCR to measure the levels of transcripts: Total RNA, 1 μg of each sample, was converted to cDNA with qScript™ cDNA SuperMix (QuantaBio, Beverly, MA, USA) for subsequent qPCR analysis of human genes. The qPCR primer pairs were: forward 5′-TTCCCGGTCACATCGAGAG-3′ and reverse 5′-TCCTGTTGCATACCGTCTAAATC-3′ for Nrf2, forward 5′-ACCACAACAGTGTGGAGAGGT-3′ and reverse 5′-CGATCCTTCGTGTCAGCAT-3′ for Keap1, forward 5′-ATGTATGACAAAGGACCCTTCC-3′ and reverse 5′-TCCCTTGCAGAGAGTACATGG-3′ for NQO1, forward 5′-AACTTTCAGAAGGGCCAGGT-3′ and reverse 5′-CTGGGCTCTCCTTGTTGC-3′ for HMOX1, forward 5′-GACAAAACACAGTTGGAACAGC-3′ and reverse 5′-CAGTCAAATCTGGTGGCATC-3′ for GCLM, forward 5′-ATATGGCAAGAAGGTGATGGTCC-3′ and reverse 5′-GGGCTTGTCCTAACAAAGCTG-3′ for TXNRD1, forward 5′-CAGGGAGGTGACTACTTCTACTC-3′ and reverse 5′-CAGGTACACCCTTAGGTCTGA-3′ for SRXN1, forward 5′-TCCAGTGTCTGTAAAGCCAGG-3′ and reverse 5′-CCAGCAGTTTTCTCTGGTTGAA-3′ for AKR1C1, forward 5′-CCCGGTCATCATTGTGGGTAA-3′ and reverse 5′-GCTTCGTGTAGGGTGTGTAGC-3′ for OSGIN1, forward 5′-GGAGCGAGATCCCTCCAAAAT-3′ and reverse 5′-GGCTGTTGTCATACTTCTCATGG-3′ for GAPDH, or forward 5′-TCAACTTTCGATGGTAGTCGCCGT-3′ and reverse 5′-TCCTTGGATGTGGTAGCCGTTTCG-3′ for 18S rRNA. The qPCR was performed with PowerUp™ SYBR™ Green Master Mix (Applied Biosystems™, Waltham, MA, USA) on a BioRad CFX96 thermal cycler and the mRNA abundance was calculated with the ΔCT method by comparing it to that of GAPDH or 18S rRNA.

ARE reporter assay for the activity of Nrf2 transcription factor: pGL4.37(*luc2P*/ARE/Hygro) vector was purchased from Promega (Madison, WI, USA). This construct contains 4 copies of ARE sequence derived from the mouse glutathione S-transferase Ya gene (mGST-ARE). To generate a second reporter under the control of the human NQO1 gene, pGL4.37 was modified by replacing the mGST-ARE sequence with 41 bp human NQO1 ARE sequence (AATCCGCAGTCACAGTGACTCAGCAGAATCTGAGCCTAGGG). For measurement of ARE activation, either pGL4.37-mGST-ARE *luc* or pGL4.37-hNQO1-ARE *luc* vector together with pCMV *Renilla* luciferase construct were transfected into HeLa cells using the Lipofectamine™ 3000 Transfection Reagent (Invitrogen, Waltham, MA, USA). At 24 h after transfection, HeLa cells were treated with L-Cystine or SFN (Santa Cruz Biotechnology, Dallas, TX, USA) for 16 h and were harvested in 1× Passive Lysis Buffer (Promega). The cell lysate was used for detecting the *Firefly* and *Renilla* luciferase activities using Dual-Glo^®^ Luciferase Assay System (Promega).

**Ubiquitylation assay:** HeLa cells were cultured in 100 mm dishes for treatment with either 0.8 mM L-Cystine or 5 μM SFN along with 10 μM MG132 for 4 h. Cells were lysed in 200 μL TBS buffer [150 mM NaCl, 10 mM Tris-HCl (pH 8.0)] containing 1% SDS and 1 mM DTT. Following 10 min boiling, soluble cell lysates were collected by centrifugation at 15,000× *g*, 4 °C for 10 min and were diluted with 800 μL TBS containing 1% Triton X-100 and 1 mM DTT. Protein A/G agarose beads (20 μL/sample, Thermo Scientific™ 78609) were added to the cell lysate by rotating at 4 °C for 1 h for removal of non-specific binding. Nrf2 antibody (1 μg, 16396-1-AP, Proteintech, Rosemont, IL, USA) was added to the cell lysate for overnight incubation at 4 °C, followed by incubation with 30 μL protein A/G agarose beads for 4 h at 4 °C. The beads were washed 4 times with TBS containing 1% Triton X-100 and 1 mM DTT. Immunoprecipitated proteins were eluted by boiling in 50 μL 1 × Laemmli sample buffer containing 5% 2-mercaptoethanol for Western blot to detect Ubiquitin.

**CRISPR-Cas9 knockout of Keap1 gene:** Keap1 CRISPR/Cas9 KO kit (sc-400190-KO-2) was purchased from Santa Cruz Biotechnology. This kit contains 3 plasmids, each encoding the Cas9 nuclease and a target-specific 20 nt guide RNA designed for maximal knockout efficiency. After transfecting into HeLa cells, GFP-positive cells were sorted by flow cytometry (BD FACSAria™ III, San Diego, CA, USA) and were seeded at single-cell density in 96-well plates for colony formation. Keap1 knockout was validated by measuring the protein using Western blot.

**ROS detection:** HeLa cells were seeded in a clear bottom 96-well dark plate and treated with 0.8 mM L-Cystine for 16 h. Following PBS wash, the cells were incubated in the dark with 10 µM 2′,7′–dichlorofluorescein diacetate (DCFDA) for 30 min at 37 °C in a 5% CO_2_ incubator. After the removal of DCFDA, 100 µL PBS was added to the cells along with H_2_O_2_ for measurement of reactive oxygen species (ROS) according to the manufacturer’s instruction (Abcam ROS detection kit, Cambridge, UK). DCF fluorescence was read at Ex485/Em535 nm by a plate reader (CLARIOstar Plus, BMG Labtech, Ortenberg, Germany).

**Cell viability and toxicity assay:** HeLa cells seeded in 96-well plates were treated with various doses of L-Cystine for 8 h before treatment with 2 μM doxorubicin for another 24 h. 3-(4,5-Dimethylthiazol-2-yl)-2,5-diphenyltetrazolium bromide (MTT, 20 µL of stock 2 mg/mL in PBS) was added to cells in 96-well plate for 2-h incubation under cell culture conditions at 37 °C in 5% CO_2_ incubator. Upon removal of the culture medium, 100 µL acidified isopropanol (53 µL of concentrated HCl in 50 mL isopropanol) was added to the cells to dissolve formazan. After 10 min of solubilization, absorbance at 570 nm was measured by a microplate reader (CLARIOstar Plus).

**Caspase activity assay for quantification of apoptosis:** Cells seeded in 24-well plates were pre-treated with L-Cystine for 8 h before treatment with 2 μM doxorubicin (Calbiochem) for 24 h. Detached cells were collected by centrifugation at 3000 rpm for 3 min and washed once with PBS before combining with corresponding adherent cells from the same well. Cells were lysed in 120 µL of lysis buffer (0.5% Nonidet P-40, 0.5 mM EDTA, 150 mM NaCl, and 50 mM Tris, pH 7.5). Caspase 3/7 activity was measured using a fluorogenic substrate Acetyl Asp-Glu-Val-Asp-7-amino-4-methylcoumarin (Ac-DEVD-AMC) [30]. Cell lysates (50 µL/well) were incubated at 37 °C for 1 h with 80 µM of synthetic peptide substrate Ac-DEVD-AMC in a total volume of 200 µL in a 96-well plate. Caspase-3/7 activity was expressed as fluorescent intensity detected at Ex365/Em450 nm by a fluorescence plate reader (CLARIOstar Plus).

**Transfection of siRNA for gene silencing:** HeLa cells grown to 80% confluence in 12-well plates were transfected with either control siRNA (sc-37007) or Nrf2 siRNA (sc-37030, Santa Cruz Biotechnology). In each reaction, 50 µL opti-MEM containing 40 pmol siRNA was mixed with 4 µL Lipofectamine 3000 and incubated for 30 min. The mixture was added to the HeLa cells, which were placed in 400 µL opti-MEM for 6-h incubation at 37 °C in a CO_2_ incubator. DMEM (500 µL) containing 20% FBS was added to cells for an 18-h incubation, followed by replacement with fresh growth medium (10% FBS in DMEM) for 48 h of culture before L-Cystine treatment.

**Statistics:** Statistical analyses were performed by GraphPad Prism 9 software. Data with one grouping variable were analyzed with one-way ANOVA and corrected by Dunnett’s multiple comparisons test. Data with two grouping variables were analyzed with two-way ANOVA and corrected by Turkey’s multiple comparisons test. Means of two groups were compared by two-tailed *t*-test. *p* value or adjusted *p* value < 0.05 was set as the threshold for significant difference.

## 3. Results

### 3.1. L-Cystine Induces Nrf2 Protein Elevation and Activates Nrf2 Transcription Factor

In addressing whether the culture medium with growth factors induces Nrf2, we discovered that changing the culture medium to fresh DMEM containing 10% FBS indeed induced Nrf2 protein in cultured HeLa cells (Figure 1A). While tracing the components, we found that fresh DMEM but not FBS induced Nrf2 elevation (Figure 1A). DMEM contains high concentrations of D-glucose (4.5 g/L) and L-glutamine (0.584 g/L or 4 mM). When adding D-glucose or L-glutamine to the cells without a medium change, there was no Nrf2 induction (Figure 1B). Induction of Nrf2 by a culture medium change was also observed in HEK293 human embryonic kidney cells, AC16 human cardiomyocytes and MCF7 human breast cancer cells (Figure 1C), implying that the phenomenon is not unique to HeLa cells. 

We further investigated the component responsible for Nrf2 induction. DMEM contains 14 amino acids or derivatives in addition to L-glutamine. Each of these amino acids was added to cells at its concentration corresponding to that in DMEM to test for Nrf2 induction (Figure 2). We found that 0.2 mM L-Cystine alone is sufficient for producing Nrf2 protein elevation (Figure 2).

To characterize the effect of L-Cystine for Nrf2 induction, we performed the dose-response curves. Since L-Cystine has low solubility in neutral pH, we dissolved it in 500 mM HCl or 200 mM NaOH to add it to the culture medium. With either method of solubilization, the elevation of Nrf2 protein started at 0.1 mM, and the degree of Nrf2 elevation increased with higher doses of L-Cystine (Figure 3A,B). The dose-response curves showed that L-Cystine at 0.8 mM was most effective for Nrf2 induction (Figure 3C). With 0.8 mM L-Cystine, time course studies revealed an induction of Nrf2 by 30 min (Figure 4A,B), which reached the peak at 4 h, before declining after 10 h (Figure 4C,D).

RNA-sequencing technology was employed to determine the transcriptomic landscape change due to L-Cystine treatment. The next generation RNA-seq of DNBseq™ technology, provided by a commercial source (BGI Genomics, San Jose, CA), revealed that 314 genes were up-regulated and 181 genes were down regulated, due to the L-Cystine treatment. The volcano mapping of differentially expressed genes revealed prominent increases of several classical Nrf2 downstream genes, including HMOX1, GCLC, GCLM, SRXN1 and Aldo-keto Reductase 1 family member C1 (AKR1C1, Figure 5A). RNA-seq data revealed that Oxidative Stress-Induced Growth Inhibitor 1 (OSGIN1) had the highest level of induction. OSGIN1 has been reported as an Nrf2 downstream gene [31]. Gene ontology biological process enrichment analysis showed that the Oxidation-Reduction Process has the highest ratio of genes altered by L-Cystine treatment (Figure 5B). The heatmap showed that most of the genes under this category were Nrf2 targets (Figure 5C).

Induction of NQO1, HMOX1, GCLM, SRXN1, TXNRD1, AKR1C1 and OSGIN1 transcripts was validated by real-time RT-PCR in a time course of L-Cystine treatment. All of these genes showed elevated transcripts at 8–24 h (Figure 6A–F). Consistent with RNA-seq data, OSGIN1 had the highest level of elevation by L-Cystine treatment (Figure 6G). L-Cystine did not cause changes in Nrf2 or Keap1 mRNA as measured by quantitative RT-PCR (Figure 6H). These data support the concept that activation of Nrf2 serves as the main molecular event with L-Cystine treatment.

Activation of the Nrf2 transcription factor was further confirmed using two different ARE reporters. We have obtained a luciferase reporter construct with the reporter gene under the control of two repeats of 41 nucleotide long promoter sequences from the mouse glutathione S-transferase Ya gene, each containing two copies of ARE sequence. ARE sequence is polymorphic. Therefore, we generated ARE reporter constructs with one copy of a 41-nucleotide-long sequence from the human NQO1 gene. Following transient transfection of the ARE reporter constructs, cells were treated with L-Cystine at various doses. With either form of the ARE reporter constructs, the detected activity of ARE is consistent with the dose-response of Nrf2 induction (Figure 7A). The level of ARE activation by L-Cystine exceeded or was comparable to that of SFN, a well-established widely used Nrf2 inducer (Figure 7A).

Induction of Nrf2 and its downstream targets by L-Cystine treatment was validated at the protein level using the Western blot. Figure 7B,C and Figure 8 show L-Cystine dose- and treatment-time-dependent elevation of the Nrf2 downstream genes, HMOX1, SRXN1, TXNRD1 and AKR1C at the protein level. As expected, Nrf2 protein elevation preceded the induction of its downstream genes (Figure 8A).

### 3.2. Mechanism of L-Cystine Induced Nrf2 Protein Elevation

L-Cystine is a disulfide conjugate of two L-Cysteine molecules and enters into cells through a cystine/glutamate antiporter encoded by the SLC7A11 gene [32]. Two inhibitors of the antiporter, erastin and sulfasalazine [33], were used to verify the dependency of the cellular entry for Nrf2 induction. Prevention of Nrf2 induction by either inhibitor confirmed that L-Cystine indeed entered into cells to induce Nrf2 (Figure 9A). Upon entering cells, L-Cystine is reduced to L-Cysteine. We treated cells in parallel, either with L-Cysteine or L-Cystine. The results showed that L-Cysteine was incapable of inducing Nrf2 (Figure 9B).

Keap1 is known as the main modulator of Nrf2 protein stability due to its binding to Nrf2 and recruitment of ubiquitin E3 ligase for Nrf2 ubiquitination. The Nrf2 protein stability measurements showed that L-Cystine treatment caused an increase in Nrf2 protein half-life from 19.4 min to 30.9 min (Figure 10A,B). The ubiquitylation assay revealed that similar to SFN, L-Cystine treatment decreased Nrf2 protein ubiquitylation (Figure 10C). Using CRISPR-Cas9 to knock out the Keap1 gene, we tested whether this abolished Nrf2 induction by L-Cystine. Among five clones of Keap1 complete knockout cells, KO1, KO2, KO3, KO4 and KO5, L-Cystine was not capable of inducing Nrf2 in four clones, KO1, KO2, KO3 and KO5 (Figure 11A,B). A summary of all five clones supports the lack of Nrf2 induction by Keap1 knockout (Figure 11C). This suggests that Nrf2 induction by L-Cystine is Keap1-dependent.

### 3.3. L-Cystine Protects against ROS Generation and Cell Death

To demonstrate that induction of Nrf2 by L-Cystine serves as a cytoprotective mechanism, we tested the inhibition of ROS generation and the anti-apoptotic capacity of L-Cystine. We found that L-Cystine reduced ROS generation by H_2_O_2_ (Figure 12A) and suppressed H_2_O_2_ from inducting caspase cleavage (Figure 12B,C). Doxorubicin is a chemotherapeutic drug that induces apoptosis [30]. HeLa cells lost approximately 60% of viability as measured by MTT assay following 24 h of exposure to 2 µM doxorubicin (Figure 13A). L-Cystine at 0.4–1.6 mM showed some protection against a loss of cell viability (Figure 13A). When caspase activity was assayed as a quantitative measure of apoptosis, inhibition of caspase activation was observed with increasing doses of L-Cystine, starting at 0.1 mM (Figure 13B). Measurements of caspase-3 cleavage confirmed the protective effect of L-Cystine (Figure 13C,D). The dependency of Nrf2 for L-Cystine-induced cytoprotection was confirmed using a siRNA to knock down Nrf2 (Figure 13E). These lines of data support that L-Cystine induces cytoprotection via Nrf2.

## 4. Discussion

This study has demonstrated that L-Cystine induces the elevation of Nrf2 protein and activates the Nrf2 transcription factor at the concentration of 0.2 mM as in the culture medium. The concentration of L-Cystine higher than that in the culture medium, e.g., 0.8–1.6 mM, induces a profound Nrf2 activation. L-Cystine-induced Nrf2 activation has been validated by RNA-seq, which demonstrated the oxidation-reduction process as a main event of L-Cystine exposure. Activation of the Nrf2 transcriptional cascade appears to be Keap1-dependent, as the Keap1 gene knockout clones failed to respond to L-Cystine for elevating Nrf2 protein. Since L-Cystine is considered a non-toxic amino acid derivative, its activation of Nrf2 provides a means for cytoprotection without concern for toxicity.

L-Cystine is a disulfide amino acid derivative, a dimer of L-Cysteine, that benefits the survival and proliferation of many cell types in cultures [34]. The reduced form of L-Cystine, i.e., L-Cysteine, is an essential amino acid capable of undergoing reversible oxidation under biological conditions, therefore serving for redox signaling and cellular control [35]. L-Cysteine enters into cells via an amino acid transporter [36]. In contrast, L-Cystine enters cells by a cystine/glutamate antiporter. Upon entering into cells, L-Cystine is reduced to L-Cysteine by cystine reductase or glutathione and thioredoxin, thereby providing the amino acid L-Cysteine essential for protein synthesis and glutathione production. The redox cycling of L-Cystine to L-Cysteine can result in protein oxidation without additional oxidants [35,37]. Because Keap1 is rich in cysteine residues, Keap1 dependency on Nrf2 induction by L-Cystine points to Keap1 oxidation as the main mechanism.

Keap1 oxidation or alkylation serves as a common molecular event leading to Nrf2 activation by small molecule Nrf2 inducers [3,5,9,10,11]. A well-established Nrf2 inducer, SFN, modifies Keap1 at cysteine residue 151 by thiocarbamylation [38,39]. The nature of SFN as an electrophile contributes to its toxicity and eliminates its clinical application, despite SFN’s popularity in cell culture and animal experiments. Two categories of Nrf2 inducers that currently have ongoing clinical trials are dimethyl fumarate and bardoxolone methyl (CDDO-Me, bardoxolone) or the related compound omaveloxolone [4,19,20]. These Nrf2 inducers are electrophiles that alkylate Keap1 at Cys 151/273/288, disabling the Keap1 checkpoint on Nrf2 [40,41]. This feature poses the possibility of off-target effects or toxicity. A Phase III clinical trial of bardoxolone for the treatment of late-stage renal failure was terminated early due to bardoxolone toxicity and increased mortality [42]. This latter result points to the urgency of developing Nrf2 inducers without safety concerns for humans.

L-Cystine fits well into the criteria for a non-toxic Nrf2 inducer. The potential limitation of utilizing L-Cystine for cytoprotection is its dependency on the presence of the cystine/glutamate antiporter for cellular uptake of L-Cystine. Many cell types contain the cystine/glutamate antiporter, a heterodimer composed of the xCT subunit of the transporter encoded by the SLC7A11 gene and ancillary 4F2hc subunit for plasma membrane localization encoded by the SLC3A2 gene [34]. Many types of cancer cells have an elevated expression of SLC7A11 and SLC3A2, a feature benefiting L-Cystine uptake and high metabolic activity for tumor growth [32,43,44]. However, not all cell types express the cystine/glutamate antiporter [34]. With cells in culture, fibroblasts, epithelial cells and macrophages are known to be efficient at L-Cystine uptake, whereas lymphocytes and stem cells are incapable of L-Cystine uptake and are therefore dependent on feeder cells to provide a source of L-Cysteine [34]. Transcriptomic profiling of gene expression among different human tissues showed higher SLC7A11 expression in the brain to a level comparable to that of cultured fibroblasts [45] (https://www.gtexportal.org/home/gene/SLC7A11 (accessed on 8 December 2022)). Proteomic profiling has revealed high levels of the cystine/glutamate transporter protein in adrenal glands, prefrontal cortex, brain, lung, heart and spinal cord [46,47]. These lines of evidence support the potential use of L-Cystine as a cytoprotective agent in those tissues.

## Figures and Tables

**Figure 1 cells-12-00291-f001:**
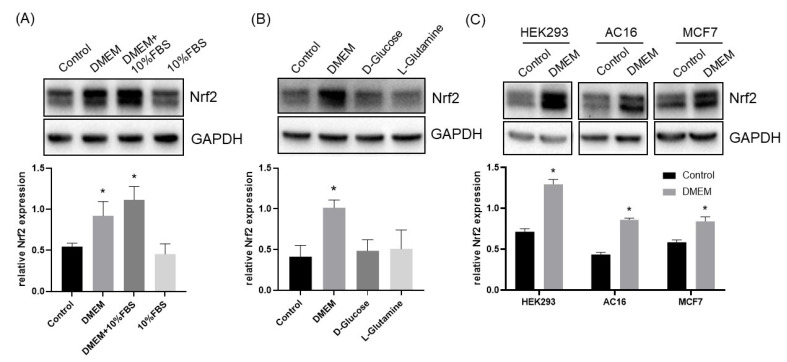
Fresh DMEM Induces Nrf2 Protein. HeLa cells at day 3 after subculture in DMEM with 10% FBS were either left untreated (control) or had a culture medium change to fresh DMEM or fresh DMEM with 10% FBS (**A**). Without changing the medium, cells were treated with 10% FBS (**A**), 25 mM glucose or 4 mM glutamine (**B**). Alternatively, HEK293, AC16 or MCF7 cells were left untreated (control) or had a culture medium change to fresh DMEM (**C**). The cells were collected at 4 h after harvesting for measuring Nrf2 protein levels by Western Blot, using GAPDH as a loading control. The intensities of the Nrf2 or GAPDH band were quantified by NIH ImageJ and presented as means ± SD of Nrf2/GAPDH ratio from three independent experiments. * indicates adjusted *p* value < 0.05 compared to the control by one-way ANOVA, corrected by Dunnett’s multiple comparisons test (**A**,**B**) or *p* < 0.05 compared to the control by student’s *t* test (**C**).

**Figure 2 cells-12-00291-f002:**
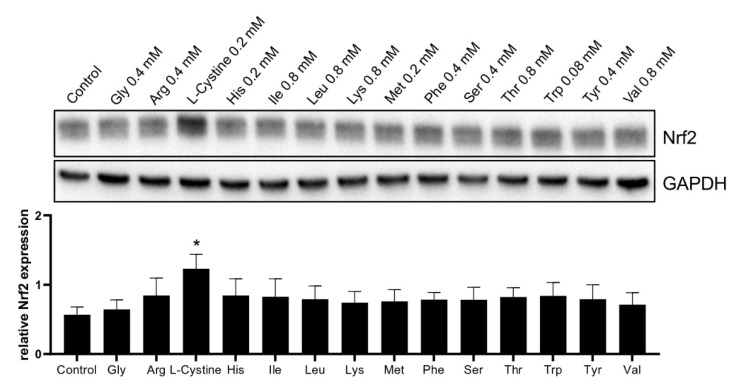
L-Cystine Induces Nrf2 Protein Elevation. The amino acid at its corresponding concentration in DMEM was added to HeLa cells at 3 days after subculture for 4 h of incubation, before harvesting for measuring Nrf2 protein levels by the Western Blot, using GAPDH as a loading control. The band intensities were quantified by NIH ImageJ to be presented as means ± SD of Nrf2/GAPDH ratio from three independent experiments. * indicates statistically significant as the adjusted *p* value < 0.05 compared to the control by one-way ANOVA, corrected by Dunnett’s multiple comparisons test.

**Figure 3 cells-12-00291-f003:**
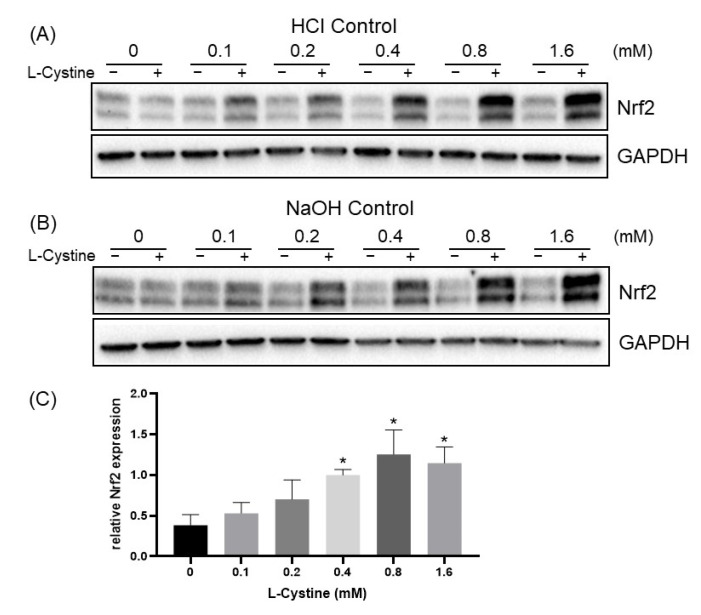
L-Cystine Dose Dependent Induction of Nrf2 Protein. L-Cystine was dissolved in either 500 mM HCl (**A,C**) or 200 mM NaOH (**B**) at a stock concentration of 0.1 M, before adding to HeLa cells on day 3 after subculture at the concentration indicated for 4 h of incubation. An equal volume of solvents was used for controls (**A**,**B**). Cells were harvested for the Western Blot to detect the levels of Nrf2 protein, using GAPDH as a loading control. The band intensities were quantified by NIH ImageJ to be presented as means ± SD of Nrf2/GAPDH ratio from three independent experiments (**C**). * indicates statistically significant compared to 0 mM L-Cystine as the adjusted *p* value < 0.05 by one-way ANOVA, corrected by Dunnett’s multiple comparisons test (**C**).

**Figure 4 cells-12-00291-f004:**
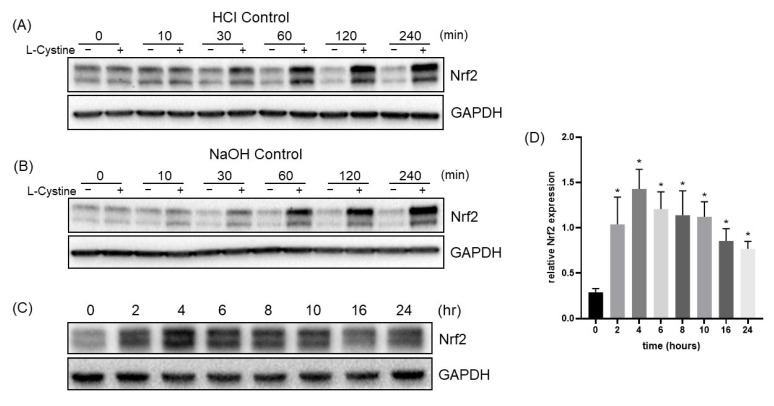
Time-Dependent Nrf2 Protein Induction by L-Cystine. HeLa cells at 3 days after subculture was treated with 0.8 mM L-Cystine from its stock dissolved in 500 mM HCL (**A**,**C**,**D**) or 200 mM NaOH (**B**) for indicated time points. An equal volume of solvents was added to the cells in the control groups. The cells were harvested at the end of the time point for Western Blot to detect the level of Nrf2 protein, using GAPDH as a loading control. The band intensities were quantified by NIH ImageJ to be presented as means ± SD of Nrf2/GAPDH ratio from three independent experiments (**D**). * indicates statistically significant compared to the 0-h time point as the adjusted *p* value < 0.05 by one-way ANOVA, corrected by Dunnett’s multiple comparisons test (**D**).

**Figure 5 cells-12-00291-f005:**
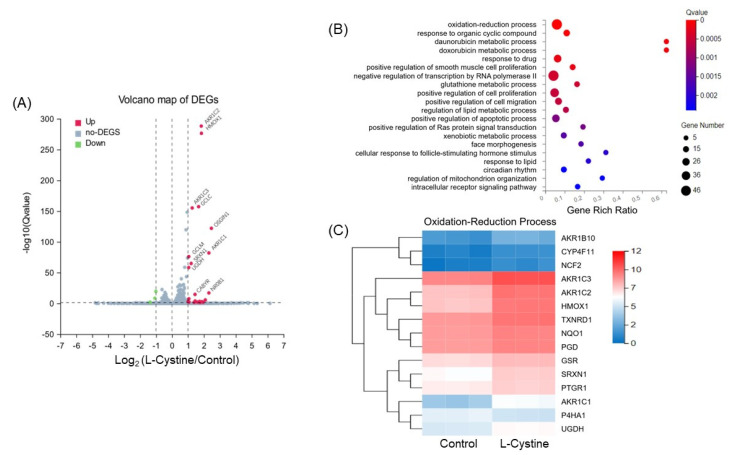
L-Cystine Induces Nrf2 Downstream Genes. HeLa cells at day 3 after subculture were treated with 0.8 mM L-Cystine for 16 h before harvesting total RNA for RNA-seq. (**A**) The volcano plot shows differentially expressed genes (DEGs) by L-Cystine treatment. Red dots indicate significantly up-regulated DEGs with −log_10_ (Q value) > 1.30 and log_2_ (fold change) > 1, which correspond to adjusted *p* value < 0.05 and fold change > 2. (**B**) Gene ontology (GO) for Biological Process Enrichment chart, showing the GO terms significantly changed by L-Cystine treatment. The X-axis is the enrichment ratio, calculated as the number of DEGs over the total number of the genes annotated to the GO term. The size of the dots reflects the ratio of DEGs annotated to the GO Term. The color represents the level of significance in Qvalue, with a threshold of ≤0.05. (**C**) Heatmap representing the variance and expression levels of up-regulated genes (log2 ≥ 0.8, adjusted *p* value ≤ 0.05) annotated in the oxidation-reduction process (GO:0055114) from the RNA-seq analyses. The data were obtained from RNA-seq of three independent sample sets.

**Figure 6 cells-12-00291-f006:**
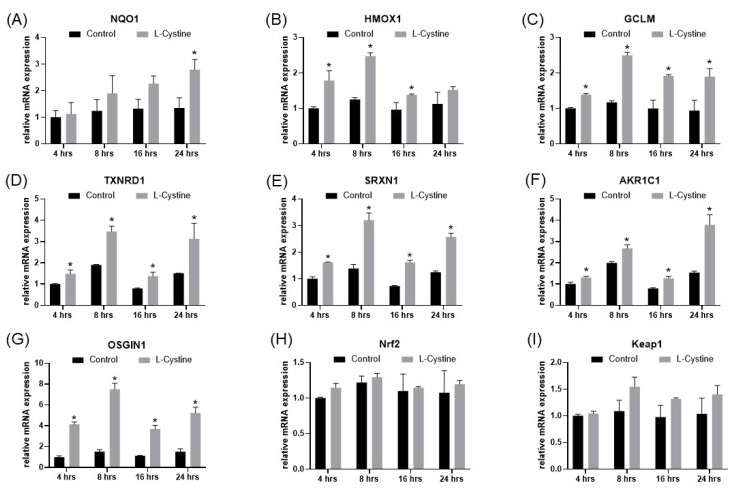
RT-PCR Measurements of Nrf2 Downstream Genes by L-Cystine Treatment. HeLa cells without or with 0.8 mM L-Cystine treatment were harvested at the indicated time for extraction of total RNA. The RNA (1 μg) was reversely transcribed into cDNA for qPCR to measure the levels of genes. The data represent means ± SD from triplicates for the abundance of the gene transcript over that of 18S rRNA (**A**–**I**). * indicates statistically significant compared to the control at the same time point as the adjusted *p* value < 0.05 by two-way ANOVA, corrected by Turkey’s multiple comparisons test (**A**–**I**).

**Figure 7 cells-12-00291-f007:**
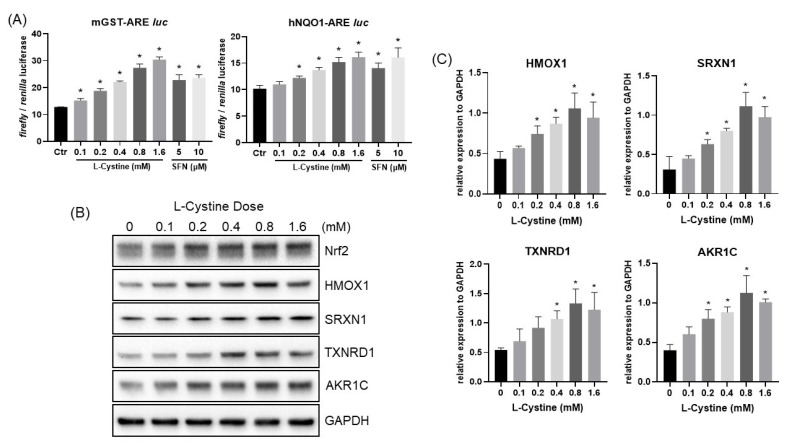
L-Cystine Dose Dependent Activation of ARE and Up-regulation of Nrf2 Downstream Genes at the Protein Level. HeLa cells were transfected with pGL4.37-mGST-ARE *luc* or pGL4.37-hNQO1-ARE *luc* vector together with pCMV of *Renilla* luciferase construct (**A**). At 48 h after transfection, cells were treated with various doses of L-Cystine or SFN as indicated for 16 h for measurements of luciferase activities. The data show means ± SD of the ratio of *Firefly*/*Renilla* luciferase from triplicate samples (**A**). Without transfection, HeLa cells were treated with L-Cystine at indicated doses for 16 h before harvesting to measure the levels of the proteins by Western Blot, using GAPDH as a loading control (**B**). Quantification of band intensities by NIH ImageJ was shown as means ± SD from three independent experiments (**C**). * indicates statistically significant compared to the control as the adjusted *p* value < 0.05 by one-way ANOVA, corrected by Dunnett’s multiple comparisons test (**A**,**C**).

**Figure 8 cells-12-00291-f008:**
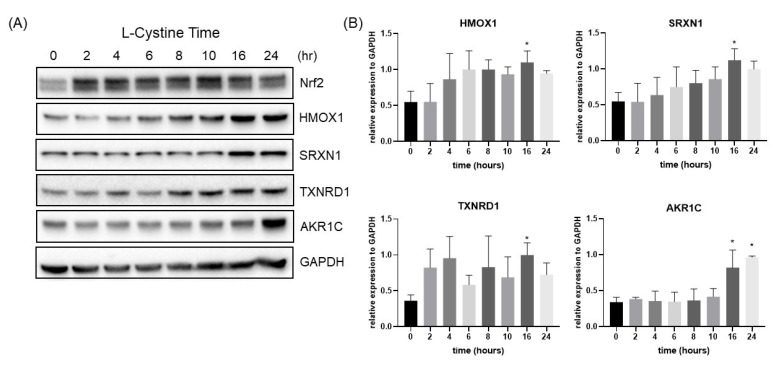
Time-Dependent Elevation of Nrf2 Downstream Genes at the Protein Level. HeLa cells were treated with 0.8 mM L-Cystine and harvested at indicated time points to measure the levels of the proteins by Western Blot, using GAPDH as a loading control (**A**). Quantification of band intensities by NIH ImageJ was shown as means ± SD from three independent experiments (**B**). * indicates statistically significant compared to the 0 h as the adjusted *p* value < 0.05 by one-way ANOVA, corrected by Dunnett’s multiple comparisons test (**B**).

**Figure 9 cells-12-00291-f009:**
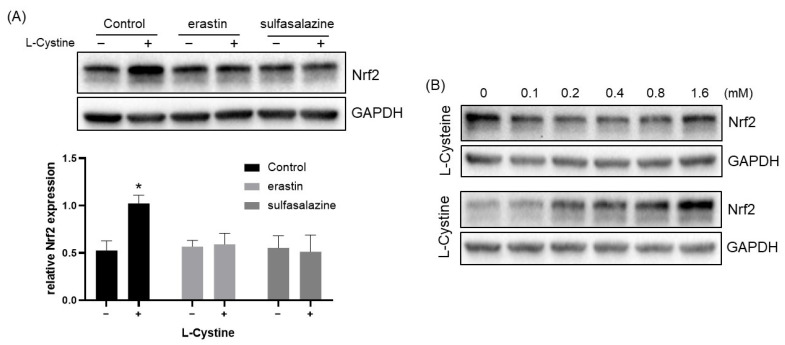
Inhibitors of Cystine/Glutamate Antiporter Block L-Cystine from Inducing Nrf2 Protein. HeLa cells at 2 days after subculture were treated with 5 μM erastin or 500 μM sulfasalazine for 24 h before treatment with 0.8 mM L-Cystine for 4 h (**A**). At day 3 after subculture, HeLa cells were treated with L-Cysteine or L-Cystine in parallel at the indicated doses for 4 h (**B**). Cells were harvested to measure Nrf2 protein levels by Western Blot, using GAPDH as a loading control. Quantification of Nrf2 over GAPDH protein band intensities by NIH ImageJ is shown as means ± SD from three independent experiments. * indicates statistically significant from the control group without L-Cystine treatment as the adjusted *p* value < 0.05 compared to the control by two-way ANOVA, corrected by Turkey’s multiple comparisons test (**A**).

**Figure 10 cells-12-00291-f010:**
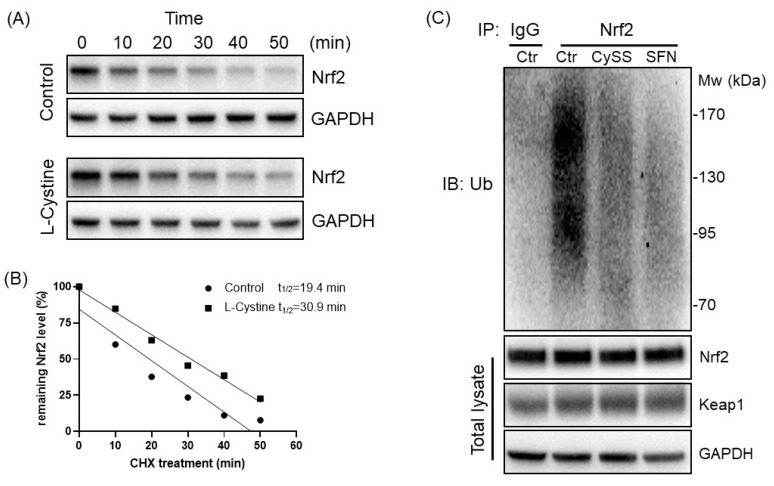
L-Cystine Extends the Half-Life of the Nrf2 Protein and decreases Nrf2 ubiquitylation. HeLa cells were treated with 0 (control) or 0.8 mM L-Cystine for 4 h followed by treatment of 100 μg/mL cycloheximide (CHX) for the indicated time. Cells were harvested at the end of the time points for detection of Nrf2 protein by Western Blot, using GAPDH as a loading control (**A**). The relative Nrf2 expression was plotted over time after adding CHX for the calculation of the half-life by linear regression (**B**). HeLa cells were treated with 0.8 mM L-Cystine (CySS) or 5 μM SFN in the presence of 10 μM MG132 for 4 h. Nrf2 protein was immunoprecipitated from cell lysate with an anti-Nrf2 antibody and resolved in 5% SDS-page gel for detection of ubiquitin by immunoblotting (**C**). Endogenous Nrf2, Keap1 and GAPDH from MG132-treated cells were included as a loading control (**C**).

**Figure 11 cells-12-00291-f011:**
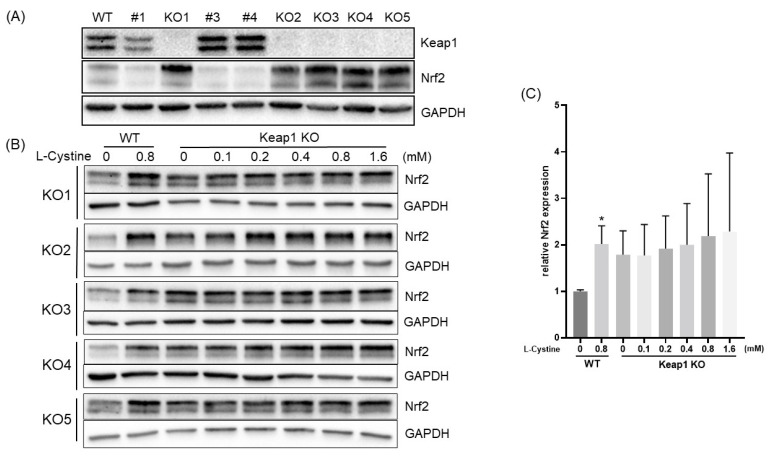
Keap1 Dependent Nrf2 Protein Induction by L-Cystine. HeLa cells were transfected with Keap1 CRISPR/Cas9 KO Plasmid for clonal selection. The clones with Keap1 knockout display a loss of Keap1 protein and elevated Nrf2 protein (**A**). The parent wild-type or Keap1 KO clones were treated with indicated doses of L-Cystine for 4 h before harvesting to measure the induction of Nrf2 protein by Western Blot, using GAPDH as a loading control (**B**). The intensities of the bands from wild-type or five clones of Keap1 KO cells were quantified using NIH ImageJ (**C**). * indicates statistically significant compared to 0 mM L-Cystine in WT cells by one-way ANOVA, corrected by Dunnett’s multiple comparisons test (**C**).

**Figure 12 cells-12-00291-f012:**
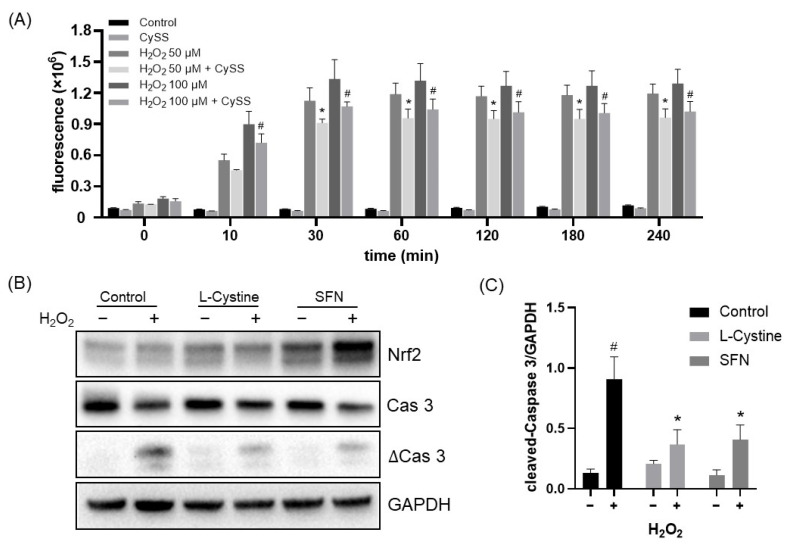
L-Cystine Reduces ROS Generation and Protects Against Oxidant-Induced Cell Death. HeLa cells were pre-treated with 0.8 mM L-Cystine (CySS) for 16 h before 30-min incubation with 2′,7′-dichlorofluorescein diacetate (DCFDA, A). H_2_O_2_ (50, 100 μM) was added to cells to induce hydroxyl radical formation (**A**). The fluorescence was recorded by a microplate reader at the time point indicated (**A**). To induce apoptosis, following 16 h of pretreatment with 0.8 mM L-Cystine or 5 μM SFN, the cells were treated with 1 mM H_2_O_2_ for 6 h before harvesting for Western blot (**B**,**C**). The data are shown as means ± SD from triplicates (**A**) or from three independent experiments by quantification of band intensities with NIH ImageJ (**C**). * or # indicates statistically significant compared to 50 or 100 μM H_2_O_2_ treatment alone, respectively, at the same time point (**A**), or compared to H_2_O_2_ treatment alone or to control without H_2_O_2_ treatment, respectively (**C**), by two-way ANOVA, corrected by Turkey’s multiple comparisons test (**A**,**C**).

**Figure 13 cells-12-00291-f013:**
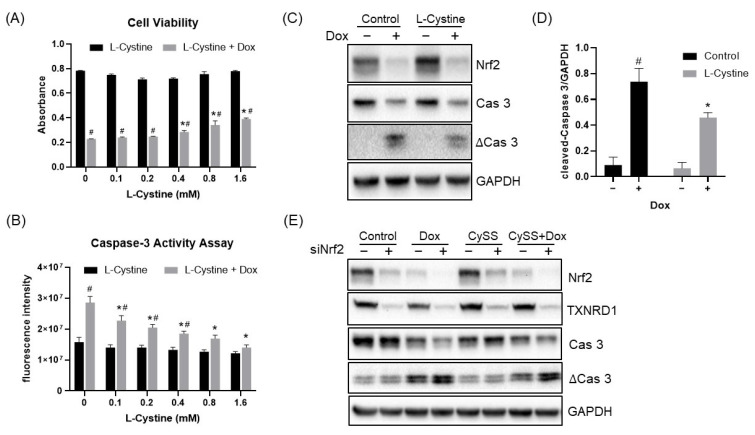
L-Cystine Protects Against Apoptosis. HeLa cells without transfection (**A**–**D**) or with transfection of control or Nrf2 siRNA for 48 h (**E**) were pre-treated with indicated doses of L-Cystine (**A**,**B**) or 0.8 mM L-Cystine (**C**–**E**) for 8 h followed by 2 μM doxorubicin (Dox) treatment for 24 h. Cells were collected for measurements of cell viability using MTT assay (A), apoptosis using caspase assay with Ac-DEVD-AMC as the substrate (**B**), caspase-3 cleavage by Western blot, using GAPDH as a loading control (**C**–**E**). Nrf2 protein and its downstream gene TXNRD1-encoded protein were measured to demonstrate the efficacy of siRNA (**D**,**E**). The data are presented as means ± SD from triplicates. * or # indicates statistically significant compared to Dox treatment alone, or the same dose of L-Cystine without Dox treatment, respectively, as the adjusted *p* value < 0.05 by two-way ANOVA, corrected by Turkey’s multiple comparisons test (**A**,**B**,**D**).

## Data Availability

The data supporting reported results are included in the article and the raw RNA-seq data are available upon request.

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
