# Peer review of "Fresh Medium or L-Cystine as an Effective Nrf2 Inducer for Cytoprotection in Cell Culture"

_cells, 2023, doi:10.3390/cells12020291_

Round 1

Reviewer 1 Report

1. General comments

This was a clear, well-written manuscript describing a logical and complete set of experiments to investigate L-cystine effects on the Nrf2 transcription factor pathway and regulation of Nrf2-dependent genes.

2. Comments

a)    This is very interesting work done by authors that have a solid understanding of the types of experiments to conduct for their project and how to interpret the results. This manuscript will likely be of interest to a wide audience.

b)    It is requested that the authors consider these two previously published works that address endogenous thiol/disulfide pairs and effects on biological redox signaling. Please consider them for citation and discussion in Section 4, in the context of the newly presented work:

Jones DP, Go YM, Anderson CL, Ziegler TR, Kinkade JM Jr, Kirlin WG. Cysteine/cystine couple is a newly recognized node in the circuitry for biologic redox signaling and control. FASEB J. 2004 Aug;18(11):1246-8. doi: 10.1096/fj.03-0971fje. Epub 2004 Jun 4. PMID: 15180957.

Ghezzi P. Oxidoreduction of protein thiols in redox regulation. Biochem Soc Trans. 2005 Dec;33(Pt 6):1378-81. doi: 10.1042/BST20051378. PMID: 16246123.

Author Response

We appreciate the enthusiasm of three reviewers.  This manuscript has been carefully revised to incorporate all comments from three reviewers.

Comments:

English language and style

( ) English very difficult to understand/incomprehensible
( ) Extensive editing of English language and style required
( ) Moderate English changes required
(x) English language and style are fine/minor spell check required
( ) I don't feel qualified to judge about the English language and style

Yes

Can be improved

Must be improved

Not applicable

Does the introduction provide sufficient background and include all relevant references?

( )

(x)

( )

( )

Are all the cited references relevant to the research?

(x)

( )

( )

( )

Is the research design appropriate?

(x)

( )

( )

( )

Are the methods adequately described?

(x)

( )

( )

( )

Are the results clearly presented?

(x)

( )

( )

( )

Are the conclusions supported by the results?

(x)

( )

( )

( )

Comments and Suggestions for Authors

Response:  The Introduction has been revised to include the initial rationale/hypothesis of the study.  The discovery is somewhat related, although not exactly, setting an example of the beauty of experimental science.

  1. General comments

This was a clear, well-written manuscript describing a logical and complete set of experiments to investigate L-cystine effects on the Nrf2 transcription factor pathway and regulation of Nrf2-dependent genes.

  1. Comments
  2. a)    This is very interesting work done by authors that have a solid understanding of the types of experiments to conduct for their project and how to interpret the results. This manuscript will likely be of interest to a wide audience.
  3. b)    It is requested that the authors consider these two previously published works that address endogenous thiol/disulfide pairs and effects on biological redox signaling. Please consider them for citation and discussion in Section 4, in the context of the newly presented work:

Jones DP, Go YM, Anderson CL, Ziegler TR, Kinkade JM Jr, Kirlin WG. Cysteine/cystine couple is a newly recognized node in the circuitry for biologic redox signaling and control. FASEB J. 2004 Aug;18(11):1246-8. doi: 10.1096/fj.03-0971fje. Epub 2004 Jun 4. PMID: 15180957.

Ghezzi P. Oxidoreduction of protein thiols in redox regulation. Biochem Soc Trans. 2005 Dec;33(Pt 6):1378-81. doi: 10.1042/BST20051378. PMID: 16246123.

Response:  These two references have been added to the Discussion (2nd paragraph).

Reviewer 2 Report

In this manuscript (ms.), the authors tried to identify Nrf2 inducers that can bypass the issues of toxicity. Despite the interesting results and novelty of this study, some comments and points which should be more resolved and/or discussed:

1.     English language should be revised. Abbreviations as well.

2.     The title of this ms. needs to be clearer; it seems that the current title is for review article not for research one. The authors should re-construct the title in a better way to recognize what is the ms. talking about.

3.     Abstract section needs to be revised and rewritten. Again, it looks like a review article.

4.     The flow of information throughout the ms. needs to be revised and re-written, if needed.

5.     The aim of this study is not clear. Please revise.

6.     Did the authors utilize a model for toxicity in vitro to evaluate the aim of this study??.. Therefor, Materials and Methods should be revised to make the study design more clearer for readers, in case of acceptance.

7.     I believe that the statistical analysis should be revised by an expert in the field.

8.     Discussion needs to be strengthened.

9.     References style should follow the instruction of Journal.

10.  The authors should provide the following at the end of the ms.:

Author Contributions:

Funding:

Informed Consent Statement:  

Author Response

We appreciate the enthusiasm of three reviewers.  This manuscript has been carefully revised to incorporate all comments from three reviewers.

Comments:

English language and style

( ) English very difficult to understand/incomprehensible
( ) Extensive editing of English language and style required
( ) Moderate English changes required
(x) English language and style are fine/minor spell check required
( ) I don't feel qualified to judge about the English language and style

Yes

Can be improved

Must be improved

Not applicable

Does the introduction provide sufficient background and include all relevant references?

( )

(x)

( )

( )

Are all the cited references relevant to the research?

( )

(x)

( )

( )

Is the research design appropriate?

( )

( )

(x)

( )

Are the methods adequately described?

( )

( )

(x)

( )

Are the results clearly presented?

(x)

( )

( )

( )

Are the conclusions supported by the results?

( )

(x)

( )

( )

Response: The manuscript has been revised with English editing from two native English speakers and experienced editors.  Reference citations have been carefully checked.  Conclusions have been checked for precision.

Comments and Suggestions for Authors

In this manuscript (ms.), the authors tried to identify Nrf2 inducers that can bypass the issues of toxicity. Despite the interesting results and novelty of this study, some comments and points which should be more resolved and/or discussed:

  1. English language should be revised. Abbreviations as well.

Response: The manuscript has been edited by two native English speakers, including Dr. Joseph S. Alpert, the Editor in Chief of the American Journal of Medicine.  A list of abbreviations has been added.  All abbreviations have been carefully checked for accuracy and consistency.

  1. The title of this ms. needs to be clearer; it seems that the current title is for review article not for research one. The authors should re-construct the title in a better way to recognize what is the ms. talking about.

Response: The title of the manuscript has been revised to incorporate this comment

  1. Abstract section needs to be revised and rewritten. Again, it looks like a review article.

Response: The abstract has been revised to incorporate this comment with added details of experimental results

  1. The flow of information throughout the ms. needs to be revised and re-written, if needed.

Response: The manuscript has been carefully revised, including the flow of information. 1) In the Results section, the first figure has been rearranged to reflect the flow of the manuscript, 2) In the Results section, cytine/glutamate antiporter inhibitor data has been moved to the section of Mechanism of L-Cystine Induced Nrf2 Protein Elevation; 3) In the Discussion, the description of the genes for cytine/glutamate antiporter has been moved to the last paragraph descripting potential implication of cystine for cytoprotection; 4) Non-relevant or not strictly related information has been removed.

  1. The aim of this study is not clear. Please revise.

Response: A paragraph has been added to the Introduction (last paragraph), to describe the aim, rationale and initial hypothesis of the study.

  1. Did the authors utilize a model for toxicity in vitro to evaluate the aim of this study??.. Therefor, Materials and Methods should be revised to make the study design more clearer for readers, in case of acceptance.

Response: The title for each section of Materials and Methods has been revised to clearly reflect the purpose of the assays, including adding the words “toxicity assay” to “Cell viability”

  1. I believe that the statistical analysis should be revised by an expert in the field.

Response:  An expert statistician has been consulted.  The descriptions of statistics in Materials and Methods have been revised.  Some of the data have been re-analyzed based on the correct statistical method, which is indicated in the figure legend. 

  1. Discussion needs to be strengthened.

Response: Discussion section has been revised, to improve the flow, and strengthen the discussion on the mechanism.

  1. References style should follow the instruction of Journal.

Response: The references have been revised to follow the style of the journal.

  1. The authors should provide the following at the end of the ms.:

Author Contributions:

Funding:

Informed Consent Statement:  

Response: These suggested sections have been added.

Reviewer 3 Report

The manuscript by Dai and Chen is well-written, easy to read and very detailed.

The experiments are sequential and well planned and performed.

Authors demonstrate that the culture conditions strongly affect the Nrf2 expression. In particular, a component of DMEM medium, the L-cystine, induces Nrf2 activity by promoting its stability and decreasing the Keap1-dependent ubiquitinylation. Additionally, the study shows that L-cystine reduces ROS production and inhibits apoptosis. The proposed mechanism for L-cystine-derived Nrf2 induction is the oxidation of Keap1, similar to that carried out by SFN or DMF but without side toxic effects (being L-cystine an essential natural aminoacid).

Just two suggestions:

1.       as GAPDH expression could be modulated by the oxidation-reduction status of the cell, at least one more house-keeping gene could be used in qRT-PCR experiments.

2.       Furthermore, given that NRF2 shows two bands in western blot, how the authors explain these two bands? and which band has been considered for the densitometry?

Author Response

We appreciate the enthusiasm of three reviewers.  This manuscript has been carefully revised to incorporate all comments from three reviewers.

English language and style

( ) English very difficult to understand/incomprehensible
( ) Extensive editing of English language and style required
(x) Moderate English changes required
( ) English language and style are fine/minor spell check required
( ) I don't feel qualified to judge about the English language and style

Yes

Can be improved

Must be improved

Not applicable

Does the introduction provide sufficient background and include all relevant references?

(x)

( )

( )

( )

Are all the cited references relevant to the research?

(x)

( )

( )

( )

Is the research design appropriate?

(x)

( )

( )

( )

Are the methods adequately described?

(x)

( )

( )

( )

Are the results clearly presented?

(x)

( )

( )

( )

Are the conclusions supported by the results?

(x)

( )

( )

( )

Comments and Suggestions for Authors

Response:  This manuscript has been edited by two native English speakers and experienced editors, including Dr. Joseph S. Alpert, the Editor-in-Chief of the American Journal of Medicine.

The manuscript by Dai and Chen is well-written, easy to read and very detailed.

The experiments are sequential and well planned and performed.

Authors demonstrate that the culture conditions strongly affect the Nrf2 expression. In particular, a component of DMEM medium, the L-cystine, induces Nrf2 activity by promoting its stability and decreasing the Keap1-dependent ubiquitinylation. Additionally, the study shows that L-cystine reduces ROS production and inhibits apoptosis. The proposed mechanism for L-cystine-derived Nrf2 induction is the oxidation of Keap1, similar to that carried out by SFN or DMF but without side toxic effects (being L-cystine an essential natural aminoacid).

Just two suggestions:

  1. as GAPDH expression could be modulated by the oxidation-reduction status of the cell, at least one more house-keeping gene could be used in qRT-PCR experiments.

Response: the reviewer is correct, we often use two different house-keeping genes for qRT-PCR.  In the revised manuscript, 18S rRNA is used for qRT-PCR experiments.  The results are consistent with that using GAPDH as the house keeping gene.

  1. Furthermore, given that NRF2 shows two bands in western blot, how the authors explain these two bands? and which band has been considered for the densitometry?

Response: NCBI genomic database shows human Nrf2 gene has 7 transcript variants, while the ENSEMBL genomic database shows 14 transcript variants of human Nrf2 gene.  This indicates the possibility of an alternative form from the well-studied 605 amino acids protein.  It is also possible that Nrf2 protein undergoes constitutive post-translational modifications.  For example, GSK3b, a constitutively active kinase, can phosphorylate Nrf2.  Most human cells show two bands of Nrf2 protein, or a doublet, by the Western blot, depending on the separation by the percentage of gels.  This phenomenon is known for many Nrf2 laboratories, which have published Western blot results of endogenous human Nrf2 as two bands or a doublet.  While the nature of such phenomenon remains to be elucidated, we used both bands as one for quantification.

Round 2

Reviewer 2 Report

Accept in present form